# Unidirectional ion transport in nanoporous carbon membranes with a hierarchical pore architecture

Lu Chen[1,2,3,10], Bin Tu[4,10], Xubin Lu [5,6,7,10✉], Fan Li [8], Lei Jiang [2,9], Markus Antonietti [3] & Kai Xiao [1,3✉]

The transport of fluids in channels with diameter of 1-2 nm exhibits many anomalous features due to the interplay of several genuinely interfacial effects. Quasi-unidirectional ion transport, reminiscent of the behavior of membrane pores in biological cells, is one phenomenon that has attracted a lot of attention in recent years, e.g., for realizing diodes for ion-conduction based electronics. Although ion rectification has been demonstrated in many asymmetric artificial nanopores, it always fails in the high-concentration range, and operates in either acidic or alkaline electrolytes but never over the whole pH range. Here we report a hierarchical pore architecture carbon membrane with a pore size gradient from 60 nm to 1.4 nm, which enables high ionic rectification ratios up to $10^4$ in different environments including high concentration neutral (3 M KCl), acidic (1 M HCl), and alkaline (1 M NaOH) electrolytes, resulting from the asymmetric energy barriers for ions transport in two directions. Additionally, light irradiation as an external energy source can reduce the energy barriers to promote ions transport bidirectionally. The anomalous ion transport together with the robust nanoporous carbon structure may find applications in membrane filtration, water desalination, and fuel cell membranes.

[1] Department of Biomedical Engineering, Southern University of Science and Technology (SUSTech), Shenzhen, China. [2] Key Laboratory of Bio-inspired Smart Interfacial Science and Technology of Ministry of Education, School of Chemistry, Beihang University, Beijing, China. [3] Department of Colloids Chemistry, Max Planck Institute of Colloids and Interfaces, Potsdam, Germany. [4] CAS Key Laboratory for Biomedical Effects of Nanomaterials and Nanosafety, CAS Center for Excellence in Nanoscience, National Center for Nanoscience and Technology, Beijing, China. [5] Gansu Province Organic Semiconductor Materials and Technology Research Center, School of Materials Science and Engineering, Lanzhou Jiaotong University, Lanzhou, China. [6] National Green Coating Equipment and Technology Research Center, Lanzhou Jiaotong University, Lanzhou, China. [7] Institute of Chemistry, Martin Luther University Halle-Wittenberg, Halle (Saale), Germany. [8] Max Planck Institute of Microstructure Physics, Halle (Saale), Germany. [9] CAS Key Laboratory of Bio-inspired Materials and Interfacial Science, Technical Institute of Physics and Chemistry, Chinese Academy of Sciences, Beijing, China. [10]These authors contributed equally: Lu Chen, Bin Tu, Xubin Lu. ✉email: xubin.lu@mail.lzjtu.cn; xiaokai@iccas.ac.cn

In biological systems, ion transport across the cell membrane is mostly directional, an embodiment of ionic rectification[1]. Unidirectional ion transport is related to an asymmetric biological nanopore structure, in which the ionic flow in one direction is almost totally suppressed[2]. This is essential for the implementation of various significant physiological functions in life processes, e.g., the regulation of cell osmotic pressure and the build-up of an action potential[3,4]. It also inspired the study of a regime of fluid mechanics under nanoscale confinement in artificial structures, named nanofluidics[5–9]. To realize such extraordinary ion transport properties in technical systems, many nanostructures based on different materials have been fabricated by various techniques, such as asymmetric nanochannels[10,11], heterogeneous membranes[12,13], and self-assembled two-dimensional materials[14–16]. The realization of unidirectional ion transport in these examples involves breaking the symmetry of the geometry, surface charge distribution, chemical composition, or channel wall wettability, separately or simultaneously[2,8,17,18].

Despite massive efforts in this field, it is still a challenge to replicate the functionality of biological nanopores and push unidirectional ion transport further for applications[7]. One reason is that ionic rectification in synthetic nanofluidic systems still shows a performance inferior to that of their natural counterparts: The rectification ratio in artificial systems is always on the order of a few hundred while biological systems almost completely suppress the ion diffusion in one direction[11,19,20]. This is because the artificial nanostructures reported in this work still fall behind their biological counterparts. Biological nanopores have an elaborate asymmetric structure with a dimension gradient of nanometer or sub-nanometer size. In contrast, artificial nanopores still fall short in the dimension gradient, asymmetric chemistry, or both[21,22]. Other weaknesses of artificial systems are that they only work in specific conditions and do not have sufficient stability to integrate these nanosized single-channel devices into macroscopic materials[10,23]. It is therefore highly desirable to develop a stable porous membrane with a nanosized asymmetric structure to overcome the abovementioned bottlenecks. From this perspective, porous carbon materials[24–27], with their controllable pore size from several angstroms to several nanometers, their chemical stability, and the available simple synthetic approaches towards them, might become the ideal materials choice to take some steps forward[16].

Here, we report a carbon membrane with a hierarchical pore architecture (CMHPA) for biomimetic, unidirectional, and stable ion transport. CMHPA has a gradient of pore sizes built up from a nanotube segment and a membrane segment and shows an ultrahigh ionic rectification ratio up to $10^4$ in different environments, including highly saline neutral, acidic, and alkaline electrolytes. The hierarchical nanostructure with a pore size gradient from 60 to 1.4 nm is thought to be the basis of the unidirectional ion transport. In addition, CMHPA is also responsive to light illumination, and photo charging can be instrumented for light-controlled ion gating for potential photoelectric energy conversion and enhanced osmosis energy harvesting. We expect that the unique ion transport phenomenon and mechanism in CMHPA have further applications, e.g. for advanced energy storage, such as the quick charging but slow discharging of supercapacitors.

## Results and discussion

**Fabrication and morphology of CMHPA.** CMHPA was fabricated by a typical chemical vapor deposition (CVD) method using asymmetric anodic aluminum oxide (AAO) membrane with a pore diameter of 100 nm (large side) and 30 nm (narrow side) as a substrate (Supplementary Table 1, Figs. 1 and 2). Figure 1a shows that the translucent pristine AAO membrane changed to a black carbon membrane with metallic gloss after CVD deposition. The carbon deposited on the AAO substrate is a slightly disordered, nitrogen (N)-doped carbon with a graphitic structure and a $pK_a$ between 2.8 and 3.5 (Supplementary Figs. 3–5). In the deposition process, the narrow side was quickly blocked, followed by the growth of a skin layer on the membrane, while the large side was still open, exposing a nanotube morphology (Fig. 1b). The thicknesses of the nanotube wall and membrane can be well controlled by tuning the chemical vapor deposition time. Figure 1c shows SEM images of the top view and bottom view of the hierarchical carbon structure. The nanotube has an opening diameter of ~60 nm, which gradually decreases to ~10 nm at the joint of the nanotube and skin layer. The thickness of the skin layer operating as a membrane segment is ~2 μm. Tiny pores exist in both the membrane segment and nanotube walls. The pore size is estimated to be ~1.4 nm by the quenched solid density functional theory (QSDFT) model with a slit/cylindrical pore shape using a nitrogen adsorption branch kernel (Fig. 1d). In this way, we know that the hierarchical carbon structure has a channel (pore) size gradient from 60 to 1.4 nm (Fig. 1e).

**Ionic rectification phenomenon of CMHPA.** The ion transport properties were measured in a homemade electrolyte cell. A porous carbon membrane with an effective testing area of $3 \times 10^4$ μm² was symmetrically placed in contact with electrolyte solutions. The transmembrane potential used in this work was stepped at 1 s per step, with a period of 21 s. For example, the scanning voltage from −1 to +1 V has a scanning step of 0.1 V per step per second. Under KCl electrolyte with various concentrations from 1 mM to 3 M, CMHPA exhibits a diode-like current–voltage response showing strong ionic rectification (Fig. 2a). The insert current under negative voltage shows that the ionic flow was completely suppressed in the concentration range from 1 mM to 1 M. A high rectification ratio ($f$) of above $10^4$ is found in the concentration range from 1 mM to 1 M. Of note, CMHPA also rectifies successfully, with $f$ up to 5000, even in high concentration electrolytes (3 M). This result is distinctly different from previous works that described ionic rectification as failing in the high-concentration range[2,19]. Due to the blocked ionic current under negative bias, the ionic current under positive voltage can be understood as electricity-driven ion transport against a concentration gradient.

To verify the universality and stability of the nanoporous carbon membrane, we further tested the ion transport under a series of electrolytes with different monovalent and bivalent cations. Typically, the porous carbon membrane shows a stable high rectification ratio of up to $10^4$ in these different electrolytes (Fig. 2b). Moreover, the high rectification ratio can be maintained even in 1 M HCl and 1 M NaOH, while $f$ decays slightly to several thousand at low concentrations (Fig. 2c, d). This breaks the common traditional experience that nanofluidic systems have ionic rectification in either acidic or alkaline electrolytes but never in both types of media[28,29].

**Mechanism of the ionic rectification phenomenon.** Electrostatic forces (range 1–100 nm) are considered an important factor in the ionic rectification phenomenon of nanofluidic systems[5]. In the CMHPA system, undoubtedly, the electric field gradient across the carbon structure resulting from the hierarchical pore architecture with a pore size gradient and a negative surface charge contributes to ultrahigh ionic rectification. This part can be qualitatively supported by a theoretical model based on the coupled Poisson and Nernst–Planck (PNP) equations (Supplementary Fig. 6). Figure 3a, b show the calculated difference in the ion accumulation area (A-area) and depletion area (D-area)

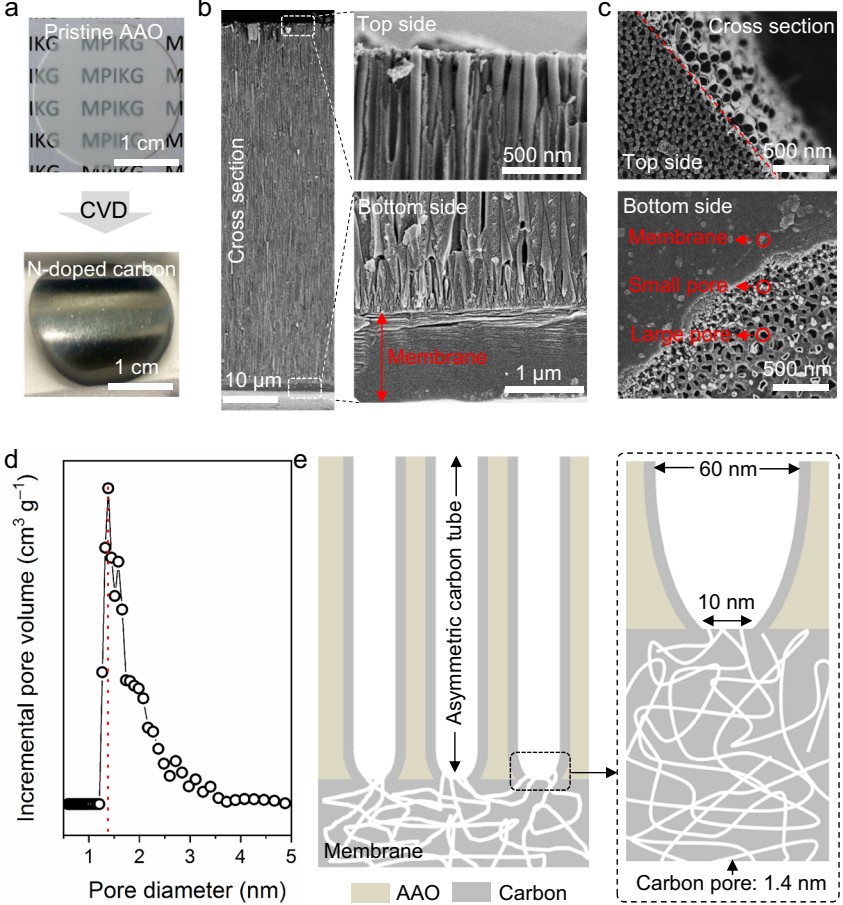

**Fig. 1 Structure and ion transport properties of CMHPA. a** Optical images of the AAO membrane before and after deposition of carbon, scale bar 1 cm. **b** Cross-section of CMHPA and typical opening view of the nanotube side and membrane side. **c** Top view of the nanotube side and bottom view of the membrane side. **d** Pore size distribution of the membrane segment. The diameter distribution peak is found at 1.4 nm. **e** Scheme illustrating CMHPA with a pore gradient: the nanotube diameter gradually changes from 60 to 10 nm, and the pore size in the membrane segment is ~1.4 nm.

under different electrical field directions, a typical concentration polarization phenomenon resulting from the ionic selectivity of CMHPA, similar to other systems[11,30,31].

Under positive voltage (Fig. 3a), ions accumulate at the nanotube and membrane junction and in the whole nanotube area but are depleted in the entrance area (bulk area). In this case, the low ion concentration can be easily compensated by ion diffusion from the bulk area, resulting in a small resistance. However, ions are depleted in the entire long nanotube under negative voltage, resulting in a long low conductivity area (i.e., large resistance). This asymmetric concentration polarization under different voltage polarities should be one reason for the high rectification ratio in CMHPA.

The above simulation results at low voltage are similar to previous work but presumably insufficient to explain the ultrahigh ionic rectification up to $10^4$ of CMHPA[10,19]. Traditionally, the electric double layers (EDLs) resulting from the electrostatic interactions between mobile ions and pore surface charges are responsible for the permselectivity of nanofluidic systems, in which the thickness is comparable to the nanopore radius somewhere along the length of the nanopore[5]. As EDLs can be screened, this mechanism has to fail at higher ion concentrations, including extreme pH values because the Debye screening length $\lambda$ is inversely proportional to the square root of the ionic strength ($\lambda \approx 0.3$ nm in a 1 M solution and $\approx 0.15$ nm in a 3 M solution)[8]. Therefore, beyond the electrostatic forces, other multiple interactions in this ~1 nm size regime[32–36], including

steric interactions (range, 0.1–2 nm) and van der Waals forces (range, 0.1–50 nm), are believed to contribute to the unique ion transport properties.

Steric interactions and van der Waals forces operate for hydrated ions at the 1 nm level, resulting in an asymmetric energy barrier for ion transport because of the gradient structure[12,37]. In the skin-layer-covered membrane with adsorbed counterions, hydrated ions are confined to form ordered chains with different molecular configurations. It is worth noting that a liquid adopts a layered structure even in the absence of interactions, simply due to geometrical constraints in a confined space[38]. To make this chain of ions slide, yield stress is needed, which depends on the specific ions but lies in all cases at approximately 0.3 V or 30 kJ mol$^{-1}$, rather typical energy for hydrated ion interactions[39]. The energy barriers for ion transport from the membrane side to the nanotube side (negative potential) are much higher than those in the opposite direction, which can be confirmed qualitatively by the calculation result under 5 V (Fig. 3c, d). A hypothetical +5 V potential is sufficient to overcome the energy barriers for ion transport. Most of the ions in the membrane segment are then pulled out of the 1.4 nm pores, resulting in an expanded depletion zone (Fig. 3c). Under the opposite bias, the energy barriers for ion transport are much higher than those under positive bias due to the asymmetric structure. A −5 V bias cannot overcome the energy barrier, i.e., a −5 V bias cannot pull the ions out of 1.4 nm pores, resulting in a similar A-area and D-area as with a −1 V bias (Fig. 3d). It must, however, be stated that ±5 V are only

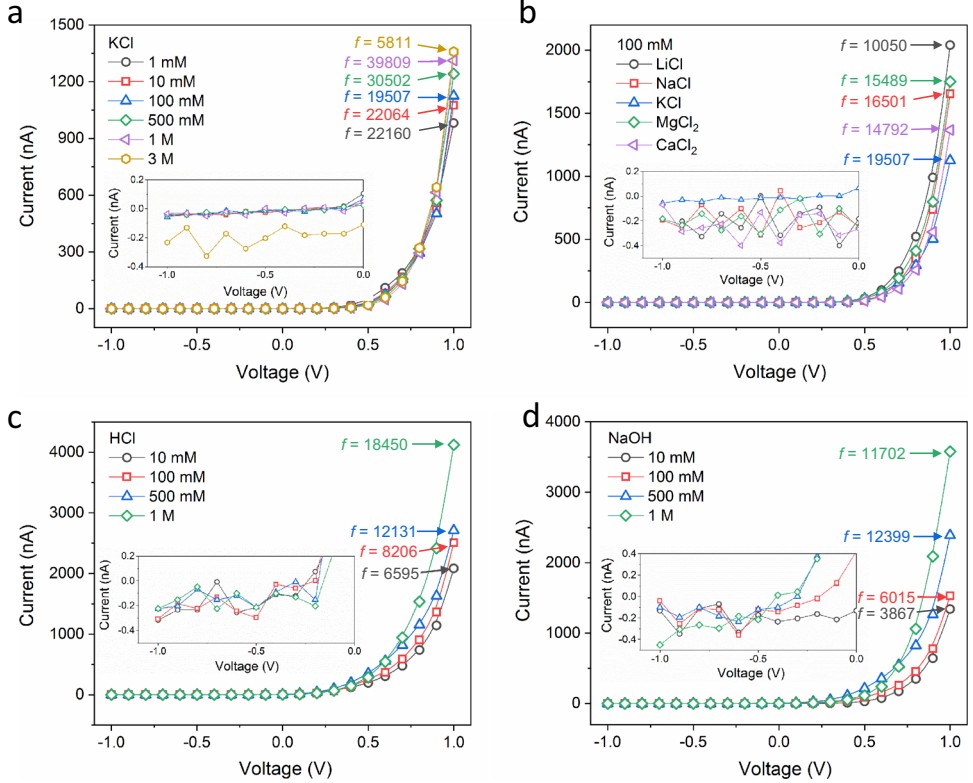

**Fig. 2 Ion transport properties and calculated rectification ratios.** Typical current–voltage curves in (**a**) KCl electrolyte with concentrations from 1 mM to 3 M; (**b**) various electrolytes with concentration of 100 mM; (**c**) HCl solutions with various concentrations from 10 mM to 1 M; and (**d**) NaOH solutions with various concentrations from 10 mM to 1 M. The numbers (f) represent the calculated rectification ratio. The measured error bar is below 1% and therefore not plotted. Error bars represent standard deviations from five independent experiments.

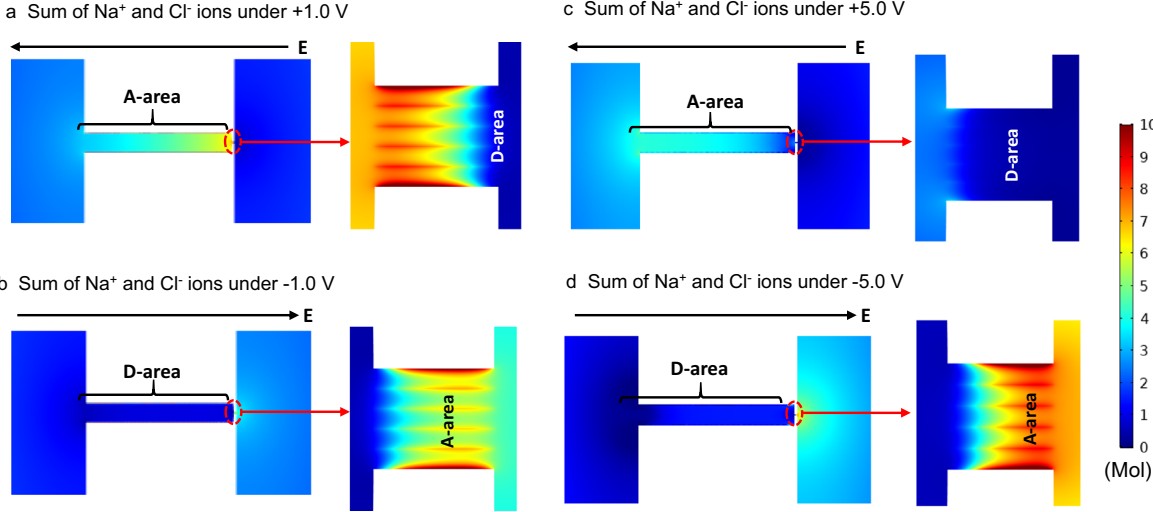

**Fig. 3 Numerical simulation of the highly rectified CMHPA. a** Ion accumulation area (A-area) and depletion area (D-area) under a +1.0 V bias. **b** Ion accumulation area and depletion area under a −1.0 V bias. **c** Ion accumulation area and depletion area under a +5.0 V bias. Under this condition, the voltage is large enough to overcome the energy barrier and pulls all ions out of the membrane segment. **d** Ion accumulation area and depletion area under a −5.0 V bias.

theoretical, as the current setup is restricted by the electrochemical window of water, i.e., water electrolysis and its overpotential.

We also studied ionic rectification by measuring the scanning voltage dependence of the ionic conductance in the present investigation. Figure 4a–d shows the current–voltage curves and calculated rectification ratio as a function of external polarization

voltage. The rectification ratio was found to be proportional to the external scanning voltage: $f = 14$ under 0.2 V, $f = 2030$ under 0.5 V, $f = 27{,}437$ under 2.0 V, and $f = 125{,}462$ under 10.0 V. The positive correlation between the rectification ratio and external voltage can be ascribed to the polarization of ions along the membrane surface, which is stronger at a higher voltage. As explained above, we know that the concentration polarizations

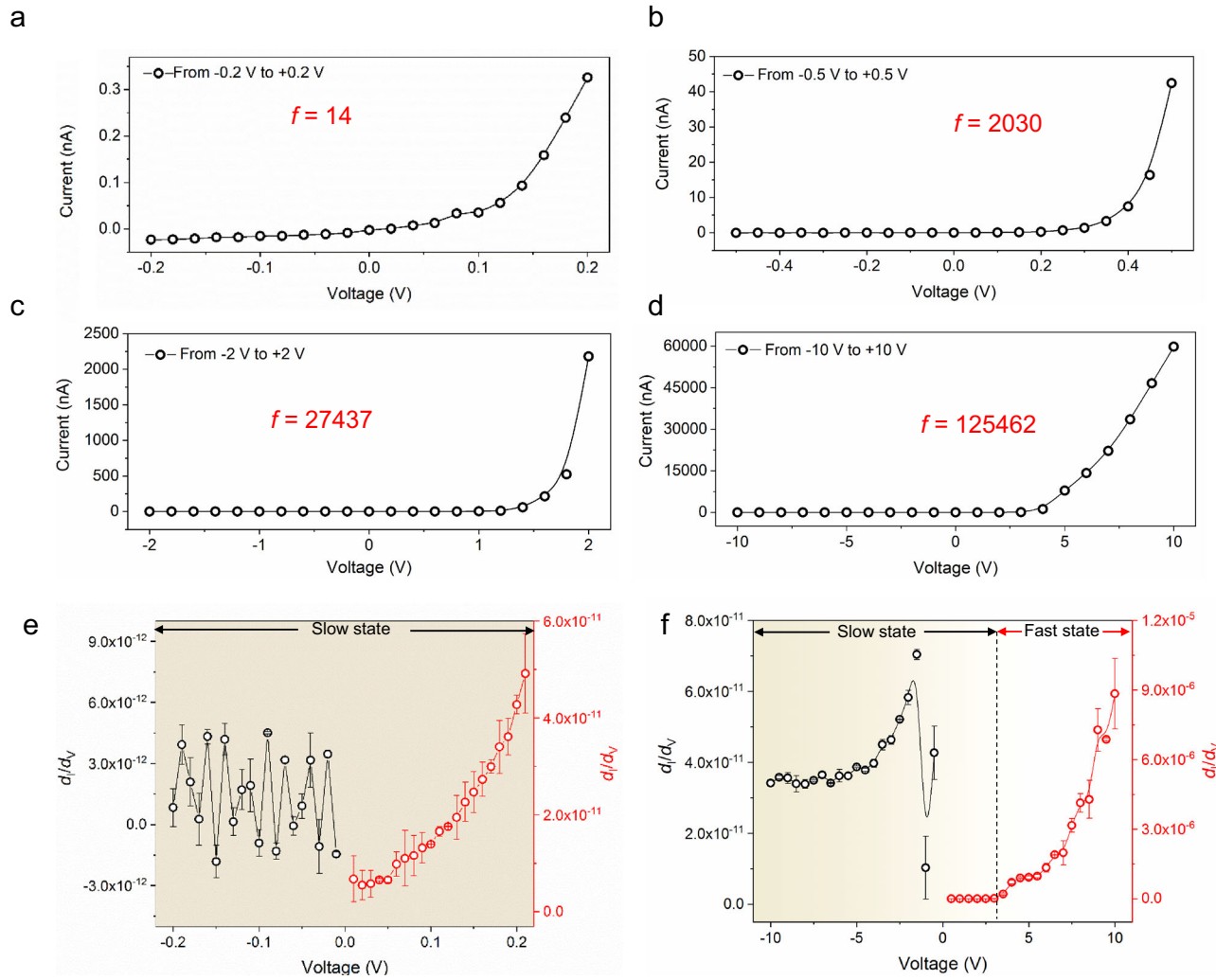

**Fig. 4 Voltage-dependent ionic rectification properties.** Ion transport properties and calculated rectification ratio at different scanning voltages (**a–d**), and calculated current growth curves $d_I/d_V$ at low voltage (from −0.2 to +0.2 V) (**e**) and high voltage (from −10 to 10 V) (**f**). The measured error bar in **a–d** is below 1% and therefore not plotted. Error bars represent standard deviations from five independent experiments.

are different at different voltage polarities. This difference will be enhanced at higher voltages, resulting in a higher rectification ratio.

From the inset drawings in Fig. 2a–d, we recognize that the ionic currents are approximately −0.2 nA under negative scanning voltage (leakage current) from −1 to 0 V. In other words, the membrane could be in a closed state under negative voltage, considering the error range of the instrument used. In fact, the low scanning voltage (Fig. 4a) proved that the membrane is still open for ion transport by the second mode of thermal ion diffusion, but at very low rates under both positive and negative voltages. With the increase of voltage from 0.5 to 10.0 V (Fig. 4b–d), the ionic currents under negative voltage and positive voltages develop the bias: CMHPA still maintains the low conductive state under high negative voltage, while the ionic current under high positive voltage increases fast. These results are more clearly presented by the calculated current growth curves $d_I/d_V$ under 0.2 and 10.0 V (Fig. 3e, f). From 0.2 to −0.2 V (Fig. 3e) and from −10 to + 3 V (Fig. 3f), the small $d_I/d_V$ indicates that the ionic diode is in a closed state. The fact that the curves were taken at different maximal voltages and thereby at different rates do not overlap and especially show different threshold voltages is typical for the dynamic character of ion polarization, i.e. the build-up of the pulling layer from continuous solution is from larger distances and thereby slow.

After reaching the threshold voltage and the corresponding ion polarization, the sharp increase in $d_I/d_V$ indicates that the ionic diode is in an open state. All these observations prove our assumption that the energy barriers in different ion transport directions are different: Under positive voltage, the energy barrier of ion transport is much lower than that under negative voltage and can be overcome by applying high external voltage[12,40].

**Photo-driven ion transport phenomenon of CMHPA.** A further interesting observation is that CMHPA shows light-responsive gating properties. As shown in Fig. 5a, these were characterized in homemade electrolyte cells with a light window, through which the carbon membrane could be illuminated. Figure 5b shows the typical current–voltage characteristics measured in the dark and under simulated solar illumination of 320 mW cm$^{-2}$. A large ionic current under positive voltage and a small ionic current under negative voltage with a rectification ratio of exceeding 6000 were observed without illumination. In contrast, both the currents under positive and negative voltages increased after illumination, with the rectification ratio dropping to ~10. In other words, the closed state, especially under negative voltage, is opened with illumination. This can be attributed to the change in surface properties of the channels induced by light, i.e., the

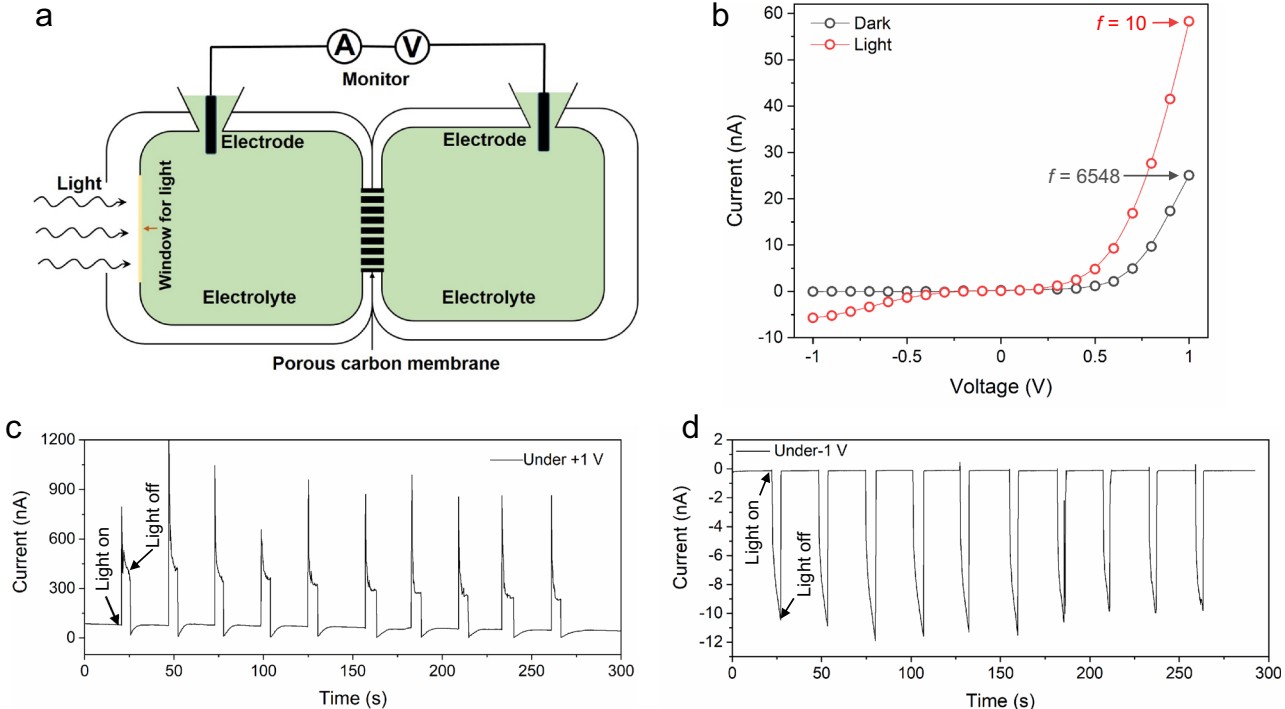

**Fig. 5 Light-gated properties. a** Schematic of light-gated ion transport of CMHPA. **b** Typical current–voltage curves and calculated rectification ratio before and after light (320 mW/cm²) irradiation at 0.01 M KCl. **c** Measured cyclic constant current under +1 V with alternating illumination. **d** Measured cyclic constant current under −1 V with alternating illumination.

negative surface charges and the coupled binding of cations are partly compensated by photoinduced holes, while the electrons stay, as long-lived hot electrons in the bulk of the material[41,42]. Such a photoinduced change in surface properties has been observed in many other carbon or semiconductor materials, e.g., graphene membranes[43,44] and carbon nitride membranes[45]. The measured cyclic constant currents under positive and negative voltages show stable and fully repeatable instant responses to illumination, proving that the light-responsive gating properties are caused by the photoelectric effect, not a photothermal effect[46]. As such, light intensity is indeed transferred in an altered ion flux, which is an artificial point of view.

Another favorable property of CMHPA is its stability. The unidirectional ion transport can be maintained over a 15 min measuring time (Supplementary Fig. 7), a key characteristic for its further application. In addition, experiments with NaOH underline that not only halide ions but also OH⁻ can be driven to transport, i.e., the system would also work as a bias membrane in an alkaline electrolyzer[47].

In summary, a CMHPA has been successfully fabricated using traditional CVD approaches. We studied its ion transport properties by investigating the voltage, concentration and illumination dependence of the ionic rectification. Our results show that the ionic rectification and ion transport characteristics of CMHPA are different from other nanofluidic systems. CMHPA showed a much higher and more stable ionic rectification ratio up to $10^4$ even in high concentration electrolyte solutions as well as in strong acid and strong base solutions, together with voltage-dependent ion transport characteristics: the rectification ratio was found to be proportional to the external scanning voltage. Furthermore, the membrane exhibited light-responsive gating properties, attributed to the change in surface properties and the coupled ion interactions induced by light.

Despite the N-doped carbon used in this work, the results suggest that other carbon materials (whether doped or not) with

negative surface charge and hierarchical pore architecture will likely also show anomalous features. Benefiting from the stable porous carbon material, these directional ion transport characteristics may find wide-ranging applications, such as in desalination membranes, osmosis energy generation devices and artificial ion pumps, as well as more complex energy conversion and storage applications, such as in ionic photovoltaic cells, directional separation membranes for alkaline electrolyzers, and supercapacitors with lower leakage currents.

## Data availability

All data generated or analyzed during this study are included in the paper and its Supplementary Information, and from the corresponding author on request.

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

## Acknowledgements

We acknowledge the support of the technicians in MPIKG. We acknowledge the help from Dr. T. Heil and A. Quetschke for TEM support, F. Syrowatka for SEM measurements. L. Chen acknowledges the support of Excellent hundred program of Beihang University. This work was financially supported by Max Planck Society and National Key Research of China.

## Author contributions

K.X. conceived and designed the experiments. X.B.L. fabricated the CMHPA. L.C. and K.X. performed the electrochemical testing and characterizations. B.T. performed the calculations. F.L and L.J. helped to analyze the data and discuss the results. L.C., K.X. and M.A. wrote the manuscript.

## Funding

## Competing interests

The authors declare no competing interests.
