## [Peer Review File · Nature Communications]

REVIEWER COMMENTS

Reviewer #1 (Remarks to the Author):

This manuscript presents a new ordered porous carbon material that shows unusual ionic rectification behavior. The device is based on the well known anodic aluminum oxide (AAO) membranes, in their asymmetric form, coated with CVD carbon. The inner nanotube-like structures are terminated by a porous carbon skin that appears to have pore sizes in the 1-2 nm size range. The coupling of the ~ 60 nm nanotube channels with the 1-2 nm pores at one end produce ionic rectification by mechanisms that are discussed and modelled in some detail. The findings seem significant, but there are issues to resolve. This review focuses on the materials-related issues, with the following significant questions / comments and then the list of minor comments / suggestions that follow.

- the role of the carbon material and its surface chemistry is not clear. Why N-doped carbon? Is the N-doping important for the claimed behavior and why?

- the authors repeatedly refer to negative surface charge, but I do not see any characterization that establishes it, like zeta potential measurements. The surface charge will almost certainly vary with pH as well, and pH dependence is an important claim in the paper.

- are we sure that the 1-2 nm pores lie only in the skin region, and not the nanotube walls (carbon coating inside the AAO)? Presumably they are measured together, on the intact devices.

- the writing and referencing create some confusion about the status of the material used (the "CMHPA") in the nanofluidic field. The current references suggest that others have studied ionic transport through these two-part structures, but the claims seem to suggest the devices are wholly new in the nanofluidics field.

- the writing overall needs improvement, if the manuscript goes further toward publication.

Minor comments / suggestions

still show a performance inferior to natural counterparts - still show performance inferior to natural counterparts

to integrate these nanosized single-channel devices - to be integrated into

tackle the describe bottlenecks - overcome the above mentioned bottlenecks.

build-up from - built-up from

hierarchical carbon structure. - delete extra space

"The pore size in the membrane segment is estimated ~ 1.4 nm by Brunauer-Emmett-Teller (BET) Surface Area Analysis (Figure 1d)." - the BET theory does not give pore sizes or pore size distributions! State the actual theory used (as currently only found in the SI).

Figure 1:

why is the membrane section in (e) shown with a foam-like structure with disconnected spherical pores??

I struggle to understand the top side in panel C. There appear to be two sub-populations of pores with different sizes (bottom-left vs. top right)

The characterization provided does not establish the claim of a gradual transition from 60 nm to 10 nm as drawn in (e). At least I do see this (unless that is what panel (c) is trying to show, in which case it needs to

be labeled and explained. I imagine this structure (the narrowing) is already known from the AAO structure and carried over to the carbon by assuming a uniform thickness of the carbon coating?

Question: how do we know the PSD relates only to the membrane section and does not include pores in the nanotube walls? The walls (coatings) seem to be thick enough to support such pores...

keeps on rectifying - slang. I believe the authors just means "also rectifies successfully"

i.e., 3.3×10^{-4} mol/m²/s. - is the m² the nominal membrane face area?

decays a little bite - decays a little bit

but never work at all pH values - "all" is a big word. I think logically the authors mean "but never work in both types of media"

Electrostatic forces (range 1–100 nm) are important for the ultrahigh ionic rectification of CMHPA.5 - I am confused by this sentence. The present paper appears to introduce the concept of a CMHPA, so how does reference 5 already know the mechanisms behind the effect for this new material?

resulting from ionic selectivity of CMHPA.30 - same comment. Suggest a wording change if this statement is about nanochannels in general, not just CMHPA.

in nanofluidic - in nanofluidics

In a 1.4 nm-sized membrane segment - ?? The segment is not 1.4 nm...

"close state" - "closed state"

along a long measuring time (Supporting Information Figure S5), a mandatory issue for its -- please give the time scale. Also, "key characteristic" would be better than "mandatory issue"

With cheap and stable porous carbon membranes, - are AAO based materials "cheap"? Even before the CVD step they would be relatively expensive. Easy to buy and convenient in the lab, yes, but potentially expensive for large scale membrane applications.

What are we looking at in Supplementary Figure 3a ??

Reviewer #2 (Remarks to the Author):

The manuscript describes ion transport properties through carbon nanoporous membranes. The membrane consists of a zone that contains parallel well-defined pores as well as porous mesh with effective pore size of ~1 nm. The topic is of potential interest to the readers of Nature Communication but additional experimental evidence is necessary to support the claims.

1. Page 4: The Authors write: "Traditionally, Debye overlap is thought to be the reason of ion selectivity or ion rectification in nanofluidic, in which the Debye screening length is comparable to the nanopore radius somewhere along the length of the nanopore." The statement is not correct, because as explained by Cervera et al. (J. Chem. Phys. 124, 104706 (2006)) and others, ion current rectification in charged and asymmetric structures depends on ion concentration in a non-monotonic manner suggesting that the maximum rectification does not occur when the complete Debye layer overlap is achieved. In fact when a complete Debye layer overlap occurs, asymmetric nanopores with positive or negative surface charges stop rectifying.

2. The manuscript unfortunately completely lacks any characterization of surface charge. The Authors assume the charge is negative but they do not explain the chemical nature of it. What is pKa of the surfaces? I am very surprised that the rectification direction remains the same in NaOH and HCl. One would expect proton adsorption in the highly acidic conditions leading to the inversed surface charge and rectification? It would be necessary to check how the pore conductance in the 'on' state depends on ion concentrations or/and find an independent way to determine what the surface charge is.

3. On page 3 the Authors write: "Due to the blocked ionic current under negative bias, the ionic current under positive voltage also can be understood as an electricity-driven ion pump." Pumping typically means transport against an electrochemical gradient, in this case it could be e.g. concentration gradient.

REVIEWER COMMENTS

Reviewer #1 (Remarks to the Author):

This manuscript presents a new ordered porous carbon material that shows unusual ionic rectification behavior. The device is based on the well known anodic aluminum oxide (AAO) membranes, in their asymmetric form, coated with CVD carbon. The inner nanotube-like structures are terminated by a porous carbon skin that appears to have pore sizes in the 1-2 nm size range. The coupling of the ~ 60 nm nanotube channels with the 1-2 pores at one end produce ionic rectification by mechanisms that are discussed and modelled in some detail. The findings seem significant, but there are issues to resolve. This review focuses on the materials-related issues, with the following significant questions / comments and then the list of minor comments / suggestions that follow.

Comments 1: - the role of the carbon material and its surface chemistry is not clear. Why N-doped carbon? Is the N-doping important for the claimed behavior and why?

Response: Thanks for your comment. The N-doped carbon we used in this work is important but not strictly necessary for a high ionic rectification ratio.

As we stated in the manuscript, there are two main factors for high ionic rectification of CMHPA. One is electrostatic force, and another one is asymmetric energy barrier. The electrostatic force is in the range of 1–100 nm and is strongly influenced by surface charge. The asymmetric energy barrier for ions transport results from multiple interactions in this ~1 nm size regime, including steric interactions (range, 0.1–2 nm) and van der Waals forces (range, 0.1–50 nm). In other words, **nanostructure** and **surface charge** are two essential factors for high ionic rectification. Other materials (or other elements doped carbon) with similar structure and surface charge with CMHPA are expected to rectify similarly.

Comments 2: - the authors repeatedly refer to negative surface charge, but I do not see any characterization that establishes it, like zeta potential measurements. The surface charge will almost certainly vary with pH as well, and pH dependence is an important claim in the paper.

Response: Thanks for your suggestion. In the revised manuscript, the zeta potential of the carbon membrane is added. The pK_a is evaluated between 2.8 and 3.5. This information is added to the revised manuscript (Page 3, line 4) and supporting information (Supplementary Figure 5).

Figure R1. Zeta-potential of CMHPA. The pK_a is estimated between 2.8 and 3.5.

Comments 3: - are we sure that the 1-2 nm pores lie only in the skin region, and not the nanotube walls (carbon coating inside the AAO)? Presumably they are measured together, on the intact devices.

Response: Thanks for your comment. Yes, you are right. The 1.4 nm pore exists both in the skin-layer membrane and the nanotube parts. This information has been added and emphasized in the revised manuscript. (Page 3, Line 10)

Comments 4:- the writing and referencing create some confusing about the status of the material used (the “CMHPA”) in the nanofluidic field. The current references suggest that others have studied ionic transport through these two-part structures, but the claims seem to suggest the devices are wholly new in the nanofluidics field.

Response: Thanks for your comment. We are sorry for the confusing language and reference. The material used in this work (CMHPA) is new, and its ion transport property has not been studied before.

However, ion transport in nanofluidics is widely studied, including some polymer heterojunction structures or inorganic heterojunction structures (Reference 13, 19, and 29). In previous work, the ionic rectification is ascribed to electrostatic forces.

In this work, the ionic rectification ratio of CMHPA is at least ten times of the previous most excellent nanofluidic system. We ascribed the superior ionic rectification phenomenon to the synthetic effect of electrostatic forces and energy barriers in the nanoconfined channel.

In the revised manuscript, the language and references are reorganized to give a clear description. (Page 4, line 1-9)

Comments 5: - the writing overall needs improvement, if the manuscript goes further toward publication.

Response: Thanks for your suggestion. The revised manuscript has been edited by the professional language editing service to improve its language.

Comments 6: Figure 1: Why is the membrane section in (e) shown with a foam-like structure with disconnected spherical pores?? I struggle to understand the top side in panel C. There appear to be two sub-populations of pores with different sizes (bottom-left vs. top right)

Response: Thanks for your suggestion. The foam-like structure in Figure 1e can't show the structure of CMHPA. In the revised manuscript, a new schematic is used. Figure 1c (top side) is the top view of the membrane. The bottom-left part is the original top view, and the top right is the top view after removing deposited carbon (or say cross-section). In the revised manuscript, the related figures are changed, and the gradual transition structures are highlighted. (Figure 1)

Figure R2. Structure and ion transport properties of CMHPA.

Comments 7: The characterization provided does not establish the claim of a gradual transition from 60 nm to 10 nm as drawn in (e). At least I do see this (unless that is what panel (c) is trying to show, in which case it needs to be labeled and explained. I imagine this structure (the narrowing) is already known from the AAO structure and carried over to the carbon by assuming a uniform thickness of the carbon coating?

Response: Thanks for your comment. Yes, you are right. The AAO substrate we used has a gradual transition structure. Figure R2 is the SEM image of the AAO substrate we used. The gradual transition structure is clear on the bottom side. This information has been added to the revised supporting information. (Supplementary Figure 2)

Figure R3. The anodic aluminium oxide (AAO) membrane with a pore diameter of 100 nm (large side) and 30 nm (narrow side) is used as a substrate.

Comments 8: how do we know the PSD relates only to the membrane section and does not include pores in the nanotube walls? The walls (coatings) seem to be thick enough to support such pores...

Response: Thanks for your comment. Yes, you are right. The 1.4 nm pore exists both in the skin-layer membrane and the nanotube parts. This information has been added and emphasized in the revised manuscript. (Page 3, Line 11)

Comments 9: Minor comments / suggestions

“The pore size in the membrane segment is estimated ~ 1.4 nm by Brunauer-Emmett-Teller (BET) Surface Area Analysis (Figure 1d).” - the BET theory does not give pore sizes or pore size distributions! State the actual theory used (as currently only found in the SI).

Response: Thanks for your suggestion. In the revised manuscript, the theory used to calculate has been added. “The pore size in the membrane segment is estimated ~ 1.4 nm by quenched solid density functional theory (QSDFT) model with slit/cylindrical pore shape using nitrogen adsorption branch kernel”. (Page 3, line 11-13)

i.e., 3.3×10^{-4} mol/m²/s. - is the m² the nominal membrane face area?

Response: Thanks for your comment. The membrane we fabricated is centimeter-level (Figure 1a). The CVD method we used is suitable for the fabrication of large-size membranes even with meter-level. To avoid misunderstanding, we delete this sentence in the revised manuscript.

Electrostatic forces (range 1–100 nm) are important for the ultrahigh ionic rectification of CMHPA.5 - I am confused by this sentence. The present paper appears to introduce the concept of a CMHPA, so how does reference 5 already know the mechanisms behind the effect for this new material?

resulting from ionic selectivity of CMHPA.30 - same comment. Suggest a wording change if this statement is about nanochannels in general, not just CMHPA.

Response: Thanks for your comment. The sentence “Electrostatic forces.....” is used to show the existing mechanism behind the reported nanofluidic systems, not for CMHPA. In the revised manuscript, this paragraph has been rephrased to avoid confusion. (Page 4, line 1-9)

With cheap and stable porous carbon membranes, - are AAO based materials “cheap”? Even before the CVD step they would be relatively expensive. Easy to buy and convenient in the lab, yes, but potential expensive for large scale membrane applications.

Response: Thanks for your comment. In the revised manuscript, the work “cheap” and other similar words have been deleted to avoid misunderstanding.

What are we looking at in Supplementary Figure 3a ??

Response: Many thanks for the comment. Supplementary Figure 3a is a silicon substrate covered by our carbon material. What we want to show is its metallic luster and smoothness in the macro dimension. In the revised manuscript, some additional notes are added to give a clear description.

still show a performance minor to natural counterparts - still show performance inferior to natural counterparts

to integrate these nanosized single-channel devices - to be integrated into

tackle the describe bottlenecks - overcome the above mentioned bottlenecks.

build-up from - built-up from

hierarchic al carbon structure. - delete extra space

keeps on rectifying - slang. I believe the authors just means “also rectifies successfully”

decays a little bite - decays a little bit

but never work at all pH values - “all” is a big word. I think logically the authors mean “but never work in both types of media”

in nanofluidic - in nanofluidics

In a 1.4 nm-sized membrane segment - ?? The segment is not 1.4 nm...

“close state” - “closed state”

along a long measuring time (Supporting Information Figure S5), a mandatory issue for its -- please give the time scale. Also, “key characteristic” would be better than “mandatory issue”

Response: Many thanks for the comments and suggestions. In the revised manuscript, these minor language comments have been solved point by point. The revised manuscript has been edited by the professional language editing service to improve its language. The confused words and figures have been corrected to give a clear description.

Reviewer #2 (Remarks to the Author):

The manuscript describes ion transport properties through carbon nanoporous membranes. The membrane

consists of a zone that contains parallel well-defined pores as well as porous mesh with effective pore size of ~ 1 nm. The topic is of potential interest to the readers of Nature Communication but additional experimental evidence is necessary to support the claims.

Comments 1: 1. Page 4: The Authors write: “Traditionally, Debye overlap is thought to be the reason of ion selectivity or ion rectification in nanofluidic, in which the Debye screening length is comparable to the nanopore radius somewhere along the length of the nanopore.” The statement is not correct, because as explained by Cervera et al. (J. Chem. Phys. 124, 104706 (2006)) and others, ion current rectification in charged and asymmetric structures depends on ion concentration in a non-monotonic manner suggesting that the maximum rectification does not occur when the complete Debye layer overlap is achieved. In fact when a complete Debye layer overlap occurs, asymmetric nanopores with positive or negative surface charges stop rectifying.

Response: Thanks for your comment and teaching. In the revised manuscript, this sentence was changed to “Traditionally, the electric double layers (EDLs) resulting from the electrostatic interactions between mobile ions and the pore surface charges are responsible for the permselectivity of nanofluidic systems”. (Page 4, line 17-20)

Comments 2: 2. The manuscript unfortunately completely lacks any characterization of surface charge. The Authors assume the charge is negative but they do not explain the chemical nature of it. What is pK_a of the surfaces? I am very surprised that the rectification direction remains the same in NaOH and HCl. One would expect proton adsorption in the highly acidic conditions leading to the inverted surface charge and rectification? It would be necessary to check how the pore conductance in the ‘on’ state depends on ion concentrations or/and find an independent way to determine what the surface charge is.

Response: Thanks for your comment. In the revised manuscript, the zeta potential of carbon membrane is added. The pK_a is evaluated between 2.8 and 3.5. (Figure R4)

Figure R4. Zeta-potential of CMHPA. The pK_a is estimated between 2.8 and 3.5.

The pore conductance under +1V (in the ‘on’ state) depends on ion concentrations is summarized as Figure R5, which is similar to biological potassium nanochannel (Phys. Rev. E 2005, 71, 021912.) but different with the surface charge-controlled nanofluidic systems (J. Am. Chem. Soc. 2014, 136, 12265-12272; J. Am. Chem. Soc. 2019, 141, 3691-3698). It means that the surface charge (electrostatic force) is not the principal factor of the high ionic rectification ratio of CMHPA.

As we stated in the manuscript, there are two main factors for the ionic rectification of CMHPA. One is electrostatic force, the range of which is 1–100 nm. The electrostatic force is strongly influenced by surface charge. Beyond the electrostatic forces, other multiple interactions in this ~ 1 nm size regime, including steric interactions (range, 0.1–2 nm) and van der Waals forces (range, 0.1–50 nm), are believed to contribute to the unique ion transport properties. Steric interactions and van der Waals forces operate for hydrated ions on the 1 nm level, resulting in an asymmetric energy barrier for ions transport because of the

gradient structure. The energy barriers for ion transport from membrane side to nanotube side are much higher than the opposite direction, resulting in the “off state” under negative potential and “on state” under positive potential. This is why the CMHPA also rectifies successfully in both NaOH and HCl.

Figure R5. The pore conductance under +1V (in the ‘on’ state) depends on ion concentrations.

Comments 3: 3. On page 3 the Authors write: “Due to the blocked ionic current under negative bias, the ionic current under positive voltage also can be understood as an electricity-driven ion pump.” Pumping typically means transport against an electrochemical gradient, in this case it could be e.g. concentration gradient.

Response: Thanks for your comment. We have corrected the description in the revised manuscript: Due to the blocked ionic current under negative bias, the ionic current under positive voltage also can be understood as electricity-driven ion transport against a concentration gradient. (Page 3, line 27-29)

REVIEWERS' COMMENTS

Reviewer #1 (Remarks to the Author):

The authors have been useful revisions. I am happy with the responses to all of my original questions except #1. In that response the authors say that "N-doping is important but not strictly necessary", but do not say why N-doping is important, and don't offer any data or other evidence. It seems the theory requires a negative surface charge, but N-doping is not normally needed to achieve negative surface charge on carbons. The pKas given suggest carboxylic groups, which are common on carbon surfaces and have pKas in this range and do not contain nitrogen.

Is it possible that the authors used a synthesis protocol for N-doped carbon for some other reason, and we (and they) just do not know if the N-doping plays a role? I suspect this is true. The system is complex enough that the only way to know if "N-doping is important" as the authors claim is to make the same device with non-N-doped carbon and compare its behavior. I don't see any such data and no relevant references.

I am happy to see this work published now, but the authors should say something about N-doping (it is a notable omission). If we don't know that N-doping is important or not, that is OK - just please say something to clarify. If the theory suggests one only needs negative zeta, then one could say that the results suggest that other carbons (whether doped or not) with negative surface charge will likely also show the behavior.

Reviewer #2 (Remarks to the Author):

The Authors addressed all my comments and I am happy to recommend this manuscript for publication

REVIEWER COMMENTS

Reviewer #1 (Remarks to the Author):

The authors have been useful revisions. I am happy with the responses to all of my original questions except #1. In that response the authors say that "N-doping is important but not strictly necessary", but do not say why N-doping is important, and don't offer any data or other evidence. It seems the theory requires a negative surface charge, but N-doping is not normally needed to achieve negative surface charge on carbons. The pKas given suggest carboxylic groups, which are common on carbon surfaces and have pKas in this range and do not contain nitrogen.

Is it possible that the authors used a synthesis protocol for N-doped carbon for some other reason, and we (and they) just do not know if the N-doping plays a role? I suspect this is true. The system is complex enough that the only way to know if "N-doping is important" as the authors claim is to make the same device with non-N-doped carbon and compare its behavior. I don't see any such data and no relevant references.

I am happy to see this work published now, but the authors should say something about N-doping (it is a notable omission). If we don't know that N-doping is important or not, that is OK - just please say something to clarify. If the theory suggests one only needs negative zeta, then one could say that the results suggest that other carbons (whether doped or not) with negative surface charge will likely also show the behavior.

Response: Many thanks for your comment and recommendation. As you said, our original purpose of preparing N-doped carbon is for catalysis. It is why doped carbon is necessary. When used for ion transport, what we are sure is that negative surface charge on carbons is important, however, to be honest, it is still unclear if "N-doping is important". As your suggestion, we added a sentence in the revised manuscript to clarify that. "Despite the N-doped carbon used in this work, the results suggest that other carbons materials (whether doped or not) with negative surface charge and hierarchical pore architecture will likely also show the anomalous features" (Page 6, line 31-33).

Reviewer #2 (Remarks to the Author):

The Authors addressed all my comments and I am happy to recommend this manuscript for publication.

Response: Many thanks for your recommendation.